# CURING "MIRACLE STEPS" IN LLM MATHEMATICAL REASONING WITH RUBRIC REWARDS

## ABSTRACT

Large language models for mathematical reasoning are typically trained with outcome-based rewards, which credit only the final answer. In our experiments, we observe that this paradigm is highly susceptible to reward hacking, leading to a substantial overestimation of a model's reasoning ability. This is evidenced by a high incidence of "false positives"—solutions that reach the correct final answer through an unsound reasoning process. Through a systematic analysis with human verification, we establish a taxonomy of these failure modes, identifying patterns like *Miracle Steps*—abrupt jumps to a correct output without a valid preceding derivation. Probing experiments suggest a strong association between these *Miracle Steps* and memorization, where the model appears to recall the answer directly rather than deriving it. To mitigate this systemic issue, we introduce the Rubric Reward Model (RRM), a process-oriented reward function that evaluates the entire reasoning trajectory against problem-specific rubrics. The generative RRM provides fine-grained, calibrated rewards (0–1) that explicitly penalize logical flaws and encourage rigorous deduction. When integrated into a reinforcement learning pipeline, RRM-based training consistently outperforms outcome-only supervision across four math benchmarks. Notably, it boosts *Verified Pass@1024* on AIME2024 from 26.7% to 62.6% and reduces the incidence of *Miracle Steps* by 71%. Our work demonstrates that rewarding the solution process is crucial for building models that are not only more accurate but also more reliable.[1]

## 1 INTRODUCTION

Reinforcement learning with verifiable rewards (RLVR) has become a prominent approach in recent LLM research, primarily due to its effectiveness in improving performance on reasoning tasks that are easily verifiable (Schulman et al., 2017; Shao et al., 2024; OpenAI, 2024; Guo et al., 2025; Chen et al., 2025). Nevertheless, this paradigm is susceptible to reward hacking, leading to undesired behaviors like unfaithful chain-of-thought (CoT) (Amodei et al., 2016; Weng, 2024; Wen et al., 2025), and an overestimation of a model's capabilities (Snell et al., 2025; Wang et al., 2025).

As depicted in Figure 1, the *Standard Pass@N* metric, which considers only the correctness of the final output, exhibits favorable scaling properties. Conversely, the

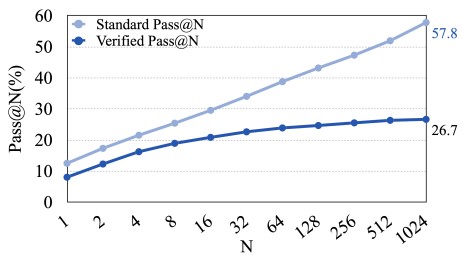

Figure 1: The *Standard Pass@N* and *Verified Pass@N* on AIME2024 for a Qwen3-4B-Base model trained with outcome-based reward (i.e. Qwen3-4B-Outcome).

*Verified Pass@N*, which scrutinizes both the reasoning process and the final answer, yields significantly lower scores and appears to plateau as N approaches 64. This significant gap between standard and verified performance motivates us to shift our focus beyond merely the errors in the reasoning process (Lightman et al., 2023; Zheng et al., 2025) to a different question:

*Why are LLMs sometimes able to arrive at the correct answer through incorrect reasoning?*

---

[1]We released our code and data at `https://anonymous.4open.science/r/Anonymous-4D6C`.

Based on this question, we conducted a preliminary human evaluation and identified several recurring patterns that lead to the correct answer through incorrect reasoning. We illustrate some of these patterns below with a concrete example. Consider the following case:

---

**Example: Question & Model's Solution & False Positive Analysis**

### Question

Determine all pairs $(f, g)$ of functions from the set of positive integers to itself that satisfy

$$f^{g(n)+1}(n) + g^{f(n)}(n) = f(n+1) - g(n+1) + 1$$

for every positive integer $n$. Here, $f^k(n)$ means $\underbrace{f(f(\ldots f(n)\ldots))}_{k}$.

Groundtruth: $\boxed{f(n) = n, g(n) = 1}$

### Model's Solution

To determine all pairs $(f, g)$ of functions from the set of positive integers to itself that satisfy the equation

$$f^{g(n)+1}(n) + g^{f(n)}(n) = f(n+1) - g(n+1) + 1$$

for every positive integer $n$, we will proceed step by step.

**Step 1: Analyze the equation for small values of $n$**
Let's start by checking the equation for $n = 1$:

$$f^{g(1)+1}(1) + g^{f(1)}(1) = f(2) - g(2) + 1$$

**Step 2: Consider simple functions**
Let's try some simple functions to see if they satisfy the equation. Suppose $f(n) = n$ and $g(n) = n$. Then the equation becomes:

$$f^{g(1)+1}(1) + g^{f(1)}(1) = f(2) - g(2) + 1$$
$$f^{1+1}(1) + g^1(1) = f(2) - g(2) + 1$$
$$f^2(1) + g(1) = f(2) - g(2) + 1$$

Since $f(n) = n$ and $g(n) = n$, we have:

$$f^2(1) = f(1) = 1, \quad g(1) = 1, \quad f(2) = 2, \quad g(2) = 2$$

So the equation becomes:

$$1 + 1 = 2 - 2 + 1 \Rightarrow 2 = 1$$

which is a contradiction. Therefore, this $(f, g)$ pair is not a solution.

**Step 3: Try another simple function**
Let's try $f(n) = n + 1$ and $g(n) = n$. Then the equation becomes:

$$f^{g(1)+1}(1) + g^{f(1)}(1) = f(2) - g(2) + 1$$

Substituting $f(n) = n + 1$ and $g(n) = n$, we get:

$$f^{1+1}(1) + g^{1+1}(1) = f(2) - g(2) + 1$$
$$f^2(1) + g^2(1) = f(2) - g(2) + 1$$

Since $f(n) = n + 1$ and $g(n) = n$, we have:

$$f^2(1) = f(f(1)) = f(2) = 3,$$
$$g^2(1) = g(g(1)) = g(1) = 1,$$
$$f(2) = 3, \quad g(2) = 2$$

So the equation becomes:

$$3 + 1 = 3 - 2 + 1 \Rightarrow 4 = 2$$

which is a contradiction. Therefore, this $(f, g)$ pair is not a solution.

**Step 4: Try a different approach**
Let's try $f(n) = n$ and $g(n) = 1$.

Then the equation becomes:

$$f^{g(1)+1}(1) + g^{f(1)}(1) = f(2) - g(2) + 1$$

Substituting $f(n) = n$ and $g(n) = 1$, we get:

$$f^{1+1}(1) + g^1(1) = f(2) - g(2) + 1$$
$$f^2(1) + g(1) = f(2) - g(2) + 1$$

Since $f(n) = n$ and $g(n) = 1$, we have:

$$f^2(1) = f(f(1)) = f(1) = 1,$$
$$g(1) = 1, \quad f(2) = 2, \quad g(2) = 1$$

So the equation becomes:

$$1 + 1 = 2 - 1 + 1 \Rightarrow 2 = 2$$

which is true. Therefore, $f(n) = n$ and $g(n) = 1$ is a solution.

**Step 5: Verify the solution**
We need to verify that $f(n) = n$ and $g(n) = 1$ is the only solution. Suppose there is another solution $(f, g)$. Then for all $n$,

$$f^{g(n)+1}(n) + g^{f(n)}(n) = f(n+1) - g(n+1) + 1$$

If $g(n) = 1$ for all $n$, then the equation becomes:

$$f^{1+1}(n) + 1^{f(n)}(n) = f(n+1) - 1 + 1$$

This simplifies to: $f^2(n) + 1 = f(n+1)$
If $f(n) = n$, then: $n^2 + 1 = n + 1$
This is not true for all $n$. Therefore, the only solution is $f(n) = n$ and $g(n) = 1$. The answer is:

$$\boxed{(f(n) = n, g(n) = 1)}$$

### False Positive Analysis

*Inductive Overgeneralization*
- Model tests $n = 1$ (passes) then claims validity for all $n$.

*Miracle Steps*
- No progress made, the model suddenly arrives at the correct pair.

*Outcome Irrelevance*
- Model miscalculates $f^2(n)$ as $n^2$ (should be $n$), but error does not affect final answer.

---

As shown in the solution part, after two failed attempts, the model suddenly performs what we term *Miracle Steps*: in Step 4, it abruptly produces the correct solution, $f(n) = n$ and $g(n) = 1$, without any valid derivation. Lacking a valid justification for its solution, the model then exhibits what we call an *Inductive Overgeneralization*: it checks only the case $n = 1$ and then directly asserts that this is the solution for all $n$. Finally, in Step 5, the model makes a calculation error, computing $f^2(n)$ as $n^2$ instead of the correct $n$, though this mistake does not affect the final answer.

These logically unsound and spurious patterns are pervasive in the model's solutions. In many cases, such patterns even enable the model to bypass the challenging steps of proof or computation and arrive at the correct final answer through an unjustified reasoning process.

Motivated by these observations, we first conduct an in-depth study to create a taxonomy of false positives in mathematical reasoning. Through a manual analysis by four annotators on the outputs of Qwen3-4B-Outcome across four benchmarks (AIME2024 (AIME, 2024), MATH500 (Hendrycks et al., 2021), AMC2023 (AMC, 2023)), and OlympiadBench (He et al., 2024), we establish a taxonomy of six distinct failure modes and identify memorization as a potential driver. We then demonstrate that this is a widespread issue by showing the prevalence of these failure modes even in state-of-the-art models, such as GPT-5 (OpenAI, 2025a) and Gemini-2.5-Pro (Comanici et al., 2025). Building on this analysis, we introduce the Rubric Reward Model (RRM), a process-oriented generative reward function grounded in problem-specific rubrics. Instead of a blunt, binary outcome

signal, the RRM assigns a fine-grained reward to the entire reasoning trace, explicitly penalizing the failure modes above and promoting step-by-step logical soundness.

We integrate this RRM into a standard reinforcement learning pipeline, training models to optimize not only for correctness but also for rigorous reasoning. Across four mathematical reasoning benchmarks, RRM-based training consistently surpasses outcome-only supervision, with especially large gains under verification metrics. For instance, on AIME2024, our method lifts *Verified Pass@1024* by 35.9 points (from 26.7 to 62.6) and narrows the Pass–Verified gap by 9.9 points (from 31.2 to 21.3). Beyond aggregate metrics, rubric-driven learning shifts the error landscape itself, reducing extreme cases such as *Miracle Steps* by 71%, demonstrating that rewarding *how* a solution is reached leads to models that are not only more accurate, but also more trustworthy in their reasoning.

## 2 RELATED WORK

**Faithful Chain-of-Thought.** LLMs can produce unfaithful CoT, misleading users (Wei et al., 2022; Anthropic, 2023a; Sharma et al., 2023; Lyu et al., 2023; Chen et al., 2024). When a model is biased towards a certain answer, it may even fabricate seemingly plausible justifications for it that are, in fact, contradictory to the facts (Turpin et al., 2023; Pacchiardi et al., 2024; Park et al., 2024; Anthropic, 2025b; Barez et al., 2025; Lam et al., 2025). This tendency can be further amplified during the feedback loop (Pan et al., 2024) and the RL process (Wen et al., 2025). Inspired by these works, we systematically investigate the patterns of unfaithful CoT in mathematical reasoning and further explore the underlying causes of this phenomenon. Building on these insights, we propose a rubric reward model to alleviate this issue and demonstrate its effectiveness.

**Rubric-Based Reward.** Rubrics have been used for reward modeling, primarily in open-ended domains lacking a single ground truth (Anthropic, 2023b; Su et al., 2025; Ma et al., 2025; Zhou et al., 2025). OpenAI utilizes specially designed rubrics to evaluate the model's capability on health (Arora et al., 2025) and AI research replication (Starace et al., 2025). Concurrently, rubric-based rewards have been applied in RL for tasks that are difficult to verify automatically, like writing, instruction-following (Viswanathan et al., 2025; Huang et al., 2025; Gunjal et al., 2025). While we adopt a similar reward mechanism, our motivation is fundamentally different. Unlike prior work using rubrics for subjective tasks, we apply them to specifically combat false positives—correct answers from flawed logic. Our rubrics are diagnostic tools derived from our taxonomy of reasoning failures, designed to penalize specific fallacies like *Miracle Steps* and enforce logical rigor.

**Outcome & Process Reward Models.** RL for mathematical reasoning typically employs Outcome Reward Models (ORMs) (Guo et al., 2025; Wei et al., 2025; Yu et al., 2025; Xu et al., 2025), which reward only the final answer, and Process Reward Models (PRMs) (Lightman et al., 2023; Wang et al., 2024; Zhang et al., 2024; He et al., 2025; Zhang et al., 2025; Zou et al., 2025), which provide step-level feedback. ORMs are a key contributor to the false positives we study, as they reward any path yielding the correct answer regardless of reasoning validity. While PRMs offer finer-grained supervision, they can be too generic to detect the subtle, high-impact fallacies prevalent in mathematical reasoning (refer to Figure 3(a)). We address this gap with the Rubric Reward Model, a problem-specific diagnostic scorer derived from our taxonomy of reasoning failures. Unlike PRMs, the RRM assigns fine-grained scores against targeted rubrics, directly penalizing patterns such as *Miracle Steps* and promoting solutions that are logically sound and verifiable.

## 3 THE FALSE POSITIVE PHENOMENON IN MATHEMATICAL REASONING

In this section, we conduct an in-depth analysis of the false positive issue. We begin by manually inspecting the outputs of Qwen3-4B-Outcome, based on which we establish a taxonomy of the observed false positives (Section 3.1). Subsequently, we design a probing experiment that suggests data leakage as a potential contributing factor (Section 3.2). Finally, we demonstrate that this issue is prevalent among other state-of-the-art LLMs, highlighting its widespread nature (Section 3.3).

### 3.1 CHARACTERIZING FALSE POSITIVES: AN EMPIRICAL TAXONOMY

To systematically characterize how models generate correct answers from flawed reasoning, we developed a taxonomy through a hybrid automated-human analysis (see Appendix C.1 for details).

Table 1: Taxonomy and distribution of false positive issues observed in Qwen3-4B-Outcome.

| Category | Description & **Example** | Count |
|---|---|---|
| Inductive Overgeneralization | The model infers a universal rule from testing a few cases (correct rule in this question), without rigorous proof. Tests $n = 1, 2, 3$ see pattern $n^2 + n$ is even, concludes "true for all $n$" (right conclusion in this question). | 21 |
| Outcome Irrelevance | The reasoning contains errors that do not affect the final answer. Computes $x = -5$ (incorrect) instead of $x = 5$ (correct), but the question asks for $\|x\|$, yielding correct value 5. | 15 |
| Neglected Operational Preconditions | The model applies algebraic or functional transformations without verifying their domains or constraints, yet the final answer remains valid coincidentally. Divides by $x$ without checking $x \neq 0$, but true solution satisfies $x = 2$ so no division-by-zero occurs. | 34 |
| Unverified Assumptions | The model introduces unproven assumptions to simplify problem solving, which happen to align with the actual extremal or target case. Assumes a triangle is equilateral to compute its area; in the given task, the maximal area case indeed corresponds to an equilateral triangle. | 18 |
| Numerical Coincidence | The derivation is logically unsound, yet due to specific numeric coincidences, the method yields the correct final number. Compute $\frac{16}{64}$, cancels out the digit '6' in the numerator and the denominator and directly arrives at $\frac{1}{4}$. | 22 |
| Miracle Steps | The solution path contains logically disconnected or invalid steps, followed unexpectedly by the correct intermediate or final expression without proper derivation. After going through some confusing steps, suddenly writes the correct $x = 1003$ with no justification. | 21 |

- We began by using Gemini-2.5-Pro to perform an initial analysis and categorization on 680 responses from 170 distinct questions, which produced a preliminary set of false positive categories. All markdown and formulas have been converted into an easily readable format.

- This automated taxonomy was then rigorously validated and refined by four expert human annotators with advanced mathematics training. The resulting human-validated framework was used to perform the quantitative analysis, revealing the model's prevalent reasoning flaws.

> During the human evaluation, we discarded several problems: (1) One problem requires an answer to be derived from the provided diagrams (see Appendix C.2.) (2) Four problems are either beyond the annotators' abilities or involve uncertainty in understanding the solution.

Table 1 details the descriptions and distribution of these false positive types observed in Qwen3-4B-Outcome's output. Six types of false positive patterns exist systematically in the model's behavior. The *Miracle Steps* category is particularly noteworthy. In these instances, the model often successfully completes a crucial step or arrives at the final answer through a process that appears logically disconnected or incomprehensible to annotators, as if miraculously bypassing the required reasoning.

## 3.2 MEMORIZATION AS A POTENTIAL CONTRIBUTOR TO FALSE POSITIVES

The prevalence of the *Miracle Steps* category motivates a critical hypothesis: **these instances may be correlated with memorization/shortcut** (Gururangan et al., 2018; Geirhos et al., 2020; Hu et al., 2024; Ye et al., 2024; Barez et al., 2025). We posit that the model, having been exposed to question-answer pairs in its training data, successfully recalls the final answer but fails to reconstruct a coherent and valid reasoning path to justify it. This failure in post-hoc rationalization manifests as a logical leap that appears miraculous to human evaluators.

To test this hypothesis, we designed a "direct answer probing" experiment. In this setup, we explicitly constrain the model to output only the final answer, forbidding any intermediate steps (refer to Figure 2 (a)). Specifically, we employ a beam search strategy to generate the Top-k answer candidates for each question and then check if the ground-truth answer is among them. The objective is to assess the model's ability to recall answers independently of its step-by-step reasoning capabilities. A high success rate in this task, particularly for questions that previously yielded *Miracle Steps*, could serve as a strong positive indicator for memorization, but it's important to note that this primarily demonstrates a correlation rather than a definitive causal relationship.

As shown in Figure 2(b), the results indicate that for a significant proportion of samples (ranging from 33% to 73% across datasets), the correct answer is found within the Top-64 candidates. These findings strongly support our memorization hypothesis. The most direct evidence comes from comparing *Miracle Steps* cases to other false positives, as shown in Figure 2(c). *Miracle Steps* problems exhibit a remarkably high answer recall rate of 83%, substantially outperforming the 63% rate for other false positive types. This disparity suggests that the "miracle" is not a leap of logic but an artifact of memory: the model successfully recalls the correct answer but fails to generate a coherent rationale for it, leading to a breakdown in the reasoning chain that is patched by the memorized result.

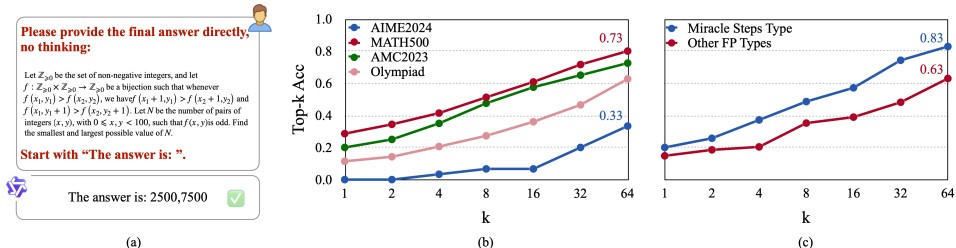

Figure 2: (a) Illustration of the direct answering setting. (b) In the direct answering setting, we report the proportion of samples from four mathematical reasoning datasets where Qwen3-4B-Outcome's answers fall within the Top-k candidates (beam search). (c) Comparison between *Miracle Steps* false positive samples and other types of false positive samples.

Table 2: False positive errors generated by the leading models on our challenge set (32 questions).

| Model | GPT-5-thinking | Gemini-2.5-Pro | Claude-4-Sonnet-thinking | o4-mini |
|---|---|---|---|---|
| *FP Rate* | 4/29 | 8/27 | 11/26 | 12/25 |

### 3.3 PREVALENCE OF FALSE POSITIVES IN STATE-OF-THE-ART MODELS

Our analysis so far has focused on a single baseline model to establish a taxonomy and a potential cause for false positives. A crucial next question is:

*Is this a systemic failure mode that affects even the most capable models?*

To answer this, we now broaden our investigation to evaluate the prevalence of these false positive phenomena across a range of state-of-the-art mathematical reasoning models. To do so, we curate a challenge set of 32 questions. These questions are selected based on a stringent criterion: for each question, our baseline model produced a correct final answer at least once across 32 attempts, yet *all* of these instances were confirmed to be false positives.

As shown in Table 2, even powerful models exhibit a non-trivial false positive rate on this challenge set: 13.8% (GPT-5), 29.6% (Gemini-2.5-Pro), 42.3% (Claude-4-Sonnet (Anthropic, 2025a)), 48% (o4-mini (OpenAI, 2025b)). This indicates that the false positive phenomenon is a systemic issue, not yet solved by scaling model size and training data alone. Appendix C.3 presents further experimental details and several concrete examples, including the specific questions and corresponding analysis.

### 3.4 EVALUATION OF GEMINI-2.5-PRO AS AN AUTOMATIC FALSE POSITIVE JUDGE

While our initial analysis relied on expert human evaluation, scaling this process requires an automated approach. To scale false positive detection beyond the human-labeled subset, we employ Gemini-2.5-Pro-0605 as an automatic judge (using the Prompt 1 in Appendix). We acknowledge that relying on an LLM introduces noise. To quantify this, we performed extensive human evaluation to assess agreement between Gemini's decisions and expert annotations.

The comprehensive evaluation results confirm Gemini's reliability: it achieves high accuracy (F1 scores: 0.90, see Table 4, 5, 6), stable performance across datasets (refer to Table 5), and no preference bias toward our rubric-based training method (refer to Table 4). Given these strengths, we adopt Gemini as a scalable, automatic false positive judge for the rest of our analysis. For detailed metrics (e.g., precision/recall scores, cross-dataset F1 values), refer to the Appendix D.1.

## 4 METHOD: TRAINING WITH RUBRIC REWARDS

The preceding analysis highlights the inadequacy of outcome-based supervision, prompting a necessary shift toward a process-oriented training paradigm. To this end, we first conduct a comparative analysis of false positive detection capabilities across three models: a process reward model, a false positive verifier, and our proposed rubric reward model (Section 4.1). Subsequently, we detail the construction process of our rubric reward model in Section 4.2.

## 4.1 Why Rubric Rewards? A Comparative Analysis

To effectively combat the false positive issue, a supervision signal must be both accurate in identifying flawed reasoning and informative enough to guide a model toward improvement. We compared three potential strategies for generating such a signal:

**(1) Process Reward Model**: This approach involves training a model on human preferences at each reasoning step. It provides step-level and trajectory-level rewards. We reuse the open-source code and model from ReasonFlux-PRM-7B (Zou et al., 2025) to compute the reward, as this model can handle responses with self-reflection steps (e.g., *1+1=3, wait...*).

**(2) False Positive Verifier**: We explicitly state the false positive categories in the prompt to Qwen3-4B (Yang et al., 2025) and ask it to determine whether the current solution has any false positive issues (see Prompt 1).

**(3) Rubric Reward Model (Ours)**: The RRM receives the question, the response, and a rubric list for this

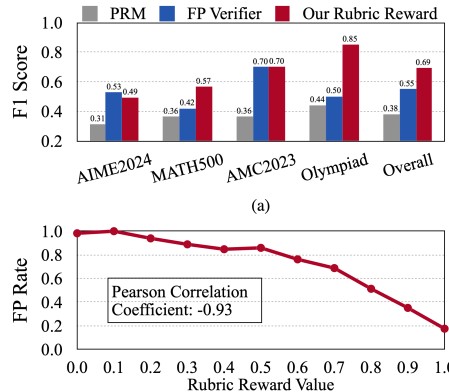

Figure 3: (a) Performance comparison of three methods for identifying false positive samples. (b) False positive rates across different rubric reward ranges.

question (more details about the RRM can be found in the next section). Given the rubric, the RRM first generates an analysis process, then assigns an integer score $s \in \{0, 1, \ldots, 10\}$ to each response. In downstream applications, this score is typically normalized to a [0, 1] range to serve as a reward. The prompt is shown in Prompt 4.

For both PRM and RRM, we need to define a false positive threshold, where any score below this value is classified as a false positive. In this experiment, the threshold is set to the value that yields the best detection performance: 1.0 for both PRM and RRM.

The results in Figure 3 show that RRM outperforms both PRM and the Verifier in two aspects:

- *Accuracy*: RRM achieves an F1 of 0.693, surpassing PRM by +0.312 and the Verifier by +0.144.
- *Continuity*: Unlike the binary Verifier and saturation-prone PRM, RRM yields fine-grained, interpretable 0–10 scores that correlate strongly with false-positive rates (98.2%→17.6% from score 0 to 10). This dense, calibrated signal rewards partially correct, fixable reasoning and penalizes errors proportionally, providing more informative gradients for training.

Overall, RRM offers both higher accuracy and richer, well-calibrated feedback, making it better suited for reducing false positives and promoting robust reasoning than PRM or binary verification.

## 4.2 Constructing the Rubric Reward Model

We build the Rubric Reward Model through a three-phase pipeline, illustrated in Figure 4. All prompts used in the entire process can be found in the Appendix A (Prompt 2-4).

**Phase 1: Rubric Synthesis.** The first step is to construct a problem-specific rubric for each training example. Our goal is to design evaluation criteria that are logically grounded and tailored to directly counteract the failure modes identified in our taxonomy (refer to Table 1). To achieve this, we prompt Gemini-2.5-Pro to generate rubrics that embody a set of core principles, thereby transforming empirical findings into actionable evaluation guidelines.

*Principle 1: Targeted principles against specific failure modes.*

- Neglected Operational Preconditions & Unverified Assumptions: Each rubric must include actionable and specific criteria. For example, instead of a vague correctness check, the rubric demands explicit verification of constraints, thereby penalizing solutions that work only coincidentally while ignoring fundamental requirements.
- Inductive Overgeneralization: We enforce the principle of completeness of sufficient conditions. The rubric must assess whether the presented evidence and reasoning are collectively sufficient

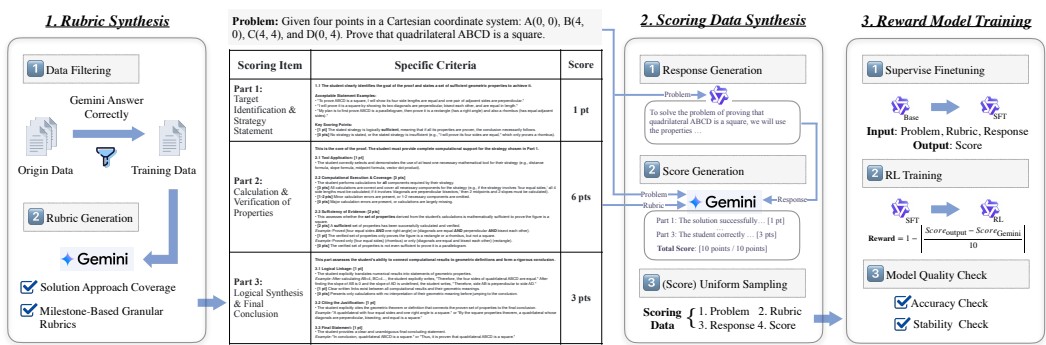

Figure 4: The pipeline of constructing our rubric reward model.

for a general proof, not merely consistent with a few examples. This shifts evaluation from pattern-matching toward requiring deductive rigor.

- Miracle Steps: The rubric mandates explicit logical linkage between steps. Any jump from confusion to an answer—without a valid derivation—fails this criterion. This ensures the reasoning chain is fully articulated, directly penalizing "miraculous" leaps symptomatic of memorization.

*Principle 2: Structure-based scaffolding.* These targeted criteria are embedded in a universal proof structure—covering strategy, computation/verification, synthesis, and conclusion. This holistic structure enables detection of broader logical flaws such as Outcome Irrelevance and Numerical Coincidence, by enforcing a coherent narrative of reasoning rather than allowing a collection of disjointed, potentially flawed calculations.

*Principle 3: Method-agnostic fairness.* All rubrics must be method-agnostic, capable of evaluating any valid solution path, not just one that matches a reference solution. This focuses the reward signal on the soundness of reasoning itself, regardless of strategy.

Based on the above principles, we carefully designed the prompt and included an illustrative, hand-crafted example in it to guide consistent generation. The detailed prompt refers to Prompt 2.

To further ensure rubric quality, we first filter out training problems for which Gemini-2.5-Pro's own solution disagrees with the reference answer, thereby eliminating problems beyond the model's capabilities and ensuring rubric feasibility. This procedure yields the dataset: $\mathcal{D}_1 = \{(\text{problem}_i, \text{rubric}_i)\}$.

**Phase 2: Scoring Data Synthesis.** Next, we generate annotated training examples for the reward model. For each $(\text{problem}_i, \text{rubric}_i)$, we produce multiple candidate responses using both the baseline model and Gemini-2.5-Pro (the latter increases the proportion of high-quality responses). We then feed the problem, rubric, and candidate response to Gemini-2.5-Pro to

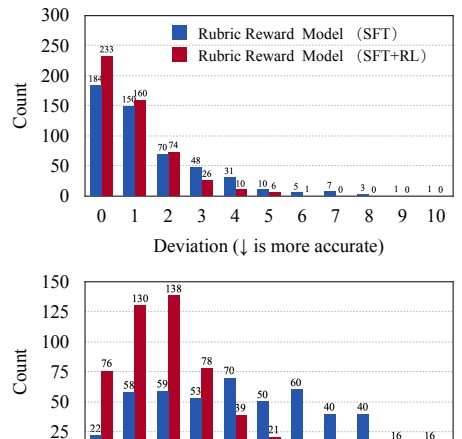

Figure 5: SFT vs. RL RRM. Accuracy: score deviation from Gemini's score; Stability: maximum variation across 5 runs, temperature set to 1.0.

obtain an integer score from 0 to 10.[2] To reduce score imbalance and avoid over-representing mid- or low-quality reasoning, we apply weighted sampling across score intervals, ensuring a more uniform distribution. After this phase, we obtain $\mathcal{D}_2 = \{(\text{problem}_i, \text{rubric}_i, \text{response}_i, \text{score}_i)\}$.

---

[2]In Appendix C.5, we have manually assessed the accuracy of Gemini's scoring. In the 1320 cases, 12 scores were higher than the actual level, and 7 scores were lower. Additionally, we have tested the stability of Gemini's scores across 5 runs, which is presented in Figure 8.

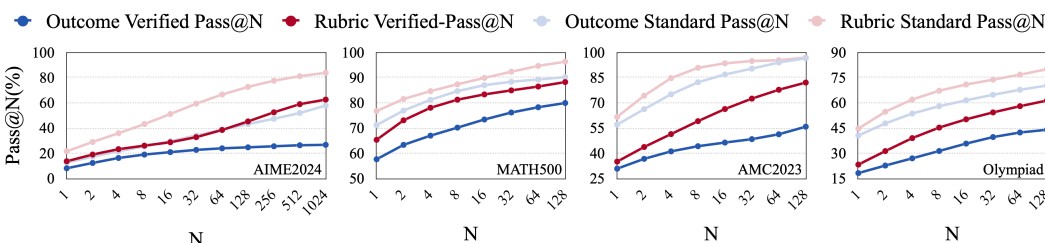

Figure 6: Performance of models trained with Outcome-Based and Rubric-Based Rewards.

**Phase 3: Reward Model Training.** We initialize our RRM from the Qwen3-4B-Base model and first perform supervised fine-tuning (SFT) on $\mathcal{D}_2$, training it to take $(\text{problem}, \text{rubric}, \text{response})$ as input and output the corresponding analysis and final score. This yields an SFT-trained checkpoint $\text{RRM}_{\text{SFT}}$. We then further refine the model using proximal policy optimization (PPO). The reward function is defined as $\text{Reward} = 1 - \left| \frac{\text{Score}_{\text{pred}} - \text{Score}_{\text{target}}}{10} \right|$. The final result, $\text{RRM}_{\text{RL}}$, serves as our rubric-aware scoring function in downstream reinforcement learning. Our rubric reward models' accuracy and stability on the hold-out test set are shown in Figure 5. Compared with $\text{RRM}_{\text{SFT}}$, $\text{RRM}_{\text{RL}}$ has significantly higher accuracy and stability. Training details refer to Appendix C.4.

## 5 EXPERIMENTS AND ANALYSIS

### 5.1 EXPERIMENTAL SETUP

**Base Model & Dataset:** We adopt Qwen3-4B-Base as the backbone model for both the baseline and our proposed approach. Training is conducted on a 9k subset of the Polaris dataset (An et al., 2025), obtained by randomly sampling 10k examples and removing examples where the provided final answer, generated by Gemini, was incorrect. We conduct evaluations on four widely used mathematical reasoning benchmarks, including AIME2024, MATH500, AMC2023, and OlympiadBench.

**Baseline & Our Method:** The baseline consists of Qwen3-4B-Base fine-tuned with PPO using a standard outcome-based reward: 1.0 for a correct final answer and 0 otherwise. The configuration is as follows: maximum sequence length of 4096 tokens, rollout size of 8, batch size of 512, learning rate of $5 \times 10^{-7}$, temperature of 1.0, and the Adam optimizer (Kingma & Ba, 2014). The training steps are set to 200 steps. We replace the outcome-based reward model in the baseline with a rubric-based reward model, while keeping all other configurations unchanged.

**Evaluation Metrics:** We use both *Standard Pass@N* and *Verified Pass@N*. For the latter, the correctness of each solution is further verified by Gemini-2.5-Pro.[3] During evaluation, solutions are generated with a temperature of 1.0 and a maximum length of 16,000 tokens.

In the main text, we focus our analysis on the 4B model. The results for the 8B model, along with comprehensive experimental details, are provided in Figure 10 and Appendix C.6, respectively.

### 5.2 MAIN RESULTS

The results in Figure 6 yield three key takeaways.

**Rubric-based rewards deliver consistent gains across datasets.** Across evaluation datasets, the rubric-trained model (pink/red) outperforms the outcome-trained model (blue) for all N under both *Standard* and *Verified Pass@N*. This pattern indicates that rewarding reasoning quality—rather than final outcomes alone—induces more generalizable problem-solving behavior.

**Gains are larger under *Verified Pass@N* and scale with N.** The improvement is notably larger for *Verified Pass@N* than for *Standard Pass@N*, and the Verified-Standard gap widens as N increases. As the candidate budget grows, the baseline tends to inflate *Standard Pass@N* by sampling more trajectories that accidentally land on the correct answer despite flawed reasoning, whereas our model produces a higher proportion of logically sound solutions. Consequently, the probability that at least one verified-correct solution appears in the N candidates grows faster for our method.

[3]A manual analysis in Table 4 confirms that Gemini-2.5-Pro does not exhibit a preference for our model's outputs over those from the baseline model, ensuring fair verification.

**Rubric rewards shrink the Verified-Standard gap.** Across all datasets and N, there is a substantial discrepancy between *Standard* and *Verified Pass@N*, underscoring the prevalence of false positives in multi-step reasoning. The gap is consistently smaller for our approach, indicating that rubric guidance suppresses spurious correctness and better aligns generation with logically valid derivations.

### 5.3 ERROR-TYPE DISTRIBUTION SHIFT AFTER RUBRIC-BASED RL

Figure 7(a) illustrates a qualitative shift: rubric-based training not only reduces the overall false positive rate but also transforms *what kinds* of false positives occur.

**Rubric rewards suppress critical errors.** The most notable effect is on the *Miracle Steps* category. Our method reduces such cases by 71% (from 175 to 50), indicating a substantial suppression of memorization-driven final-answer recalls without valid reasoning. Large reductions are also observed in other high-impact failure modes: *Neglected Operational Preconditions* (from 232 to 113) and *Unverified Assumptions* (from 213 to 167). These decreases confirm that the RRM is effective at detecting—and thereby discouraging—critical lapses in rigor.

**More detailed reasoning with minor flaws as a side effect.** Interestingly, some categories increase in frequency, notably *Outcome Irrelevance* (from 67 to 118). We view this not as regression, but as a side effect of a detailed reasoning process: by encouraging models to attempt complete, step-by-step derivations (including verification steps), we increase the chance of minor, localized mistakes arising inside an otherwise coherent reasoning chain. This effect aligns with Figure 7(b), which shows that rubric-based training encourages the model to generate more detailed and explicit reasoning steps, resulting in longer outputs. While not all added verbosity is productive, it reflects the model's attempt to build a complete logical chain, a behavior directly incentivized by the rubric. In such cases, the final answer remains correct, and the error occurs in a secondary verification or auxiliary computation (see Appendix B for an example).

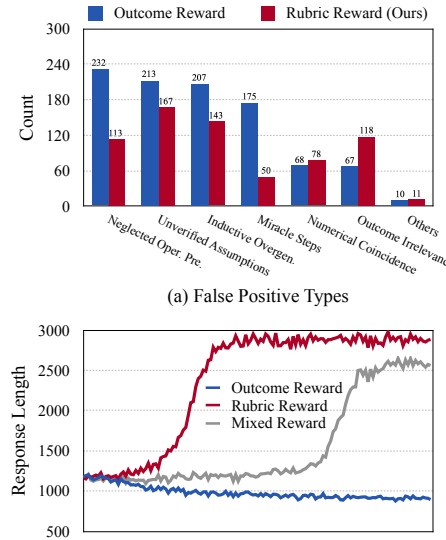

(a) False Positive Types

(b) Training Steps

Figure 7: (a) False positive distribution of two models. (b) The change in response length during RL training. "Mixed reward" means 3/4 of the rubric reward + 1/4 of the outcome reward.

## 6 LIMITATIONS AND CONCLUSION

**Limitations.** There are several limitations in our work: *(1) Dependence on strong external models.* Rubric construction relies on high-capacity models and manual filtering, limiting scalability to tasks beyond current LLM capabilities. *(2) Static reward model during RL.* The RRM is fixed after offline training; as the policy improves, the static scorer may misalign and undervalue novel yet valid reasoning. *(3) Domain and causality limitations.* Experiments are limited to mathematics, and the link between *Miracle Steps* and memorization remains indirect without full training-data provenance. Future research could address these limitations by: automating rubric synthesis to reduce manual effort, for example, via multi-agent systems; developing adaptive reward models that co-evolve with the policy to maintain alignment; and extending our analysis to other domains, like coding.

**Conclusion.** This work systematically exposes the "false positive" phenomenon in mathematical LLMs, where outcome-based rewards mask flawed reasoning. We developed a taxonomy of these failures and introduced the Rubric Reward Model to address this systemic issue. The RRM is a process-oriented reward function that provides fine-grained, calibrated scores on entire reasoning traces, directly penalizing logical fallacies. When integrated into a reinforcement learning pipeline, RRM-based training consistently and substantially outperforms outcome-only supervision. Our results provide a clear mandate: to build genuinely reliable and accurate reasoning models, we must shift our focus from validating final answers to verifying the reasoning process itself.

## REPRODUCIBILITY STATEMENT

To ensure the reproducibility of our findings, we provide a comprehensive description of our methodology, datasets, models, and evaluation procedures. All code, data, and experimental scripts have been made available at the anonymous repository: `https://anonymous.4open.science/r/Anonymous-4D6C`.

Our training process utilizes a 9k subset of the public Polaris dataset. The base models for our experiments, Qwen3-4B-Base and Qwen3-8B-Base, are publicly available open-source models. Detailed hyperparameters for the PPO training of both the outcome-based baseline and our rubric-based model are provided in Appendix C.6. This includes learning rates, batch sizes, and rollout configurations. The hardware setup (8x NVIDIA A800-80G GPUs) is also specified. The generation parameters for evaluation (e.g., temperature, max tokens) are documented in Section 5.

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

## THE USE OF LARGE LANGUAGE MODELS

LLMs were employed in a limited capacity for writing optimization. Specifically, the authors provided their own draft text to the LLM, which in turn suggested improvements such as corrections of grammatical errors, clearer phrasing, and removal of non-academic expressions. LLMs were also used to inspire possible titles for the paper. While the system provided suggestions, the final title was decided and refined by the authors and is not directly taken from any single LLM output. In addition, LLMs were used as coding assistants during the implementation phase. They provided code completion and debugging suggestions, but all final implementations, experimental design, and validation were carried out and verified by the authors. Importantly, LLMs were **NOT** used for generating research ideas, designing experiments, or searching and reviewing related work. All conceptual contributions and experimental designs were fully conceived and executed by the authors.

# A    PROMPTS

---

## Prompt 1: Gemini's False Positive Detection

You will receive the following three items: (1) A math problem; (2) A standard answer; (3) A student's submitted answer (including their problem-solving process and final answer).

Your task is: (1) Carefully review the student's problem-solving process; (2) Determine whether there are errors, logical flaws, or imprecise points in the method used to arrive at the final answer; (3) If there are problems, explain the type of error and elaborate on why the correct answer was still obtained under such circumstances; (4) The problem-solving process may contain some self-corrected errors, e.g., "1+1=3 wait, 1+1=2" – these are not considered errors but rather the model's thinking process before finding the correct method and answer; (5) There are six types of errors in total. If there are any beyond these six, please explain them additionally.

```
1. Inductive Overgeneralization (overgeneralization/incomplete induction/insufficient enumeration)
     - Typical symptoms:
          - Asserting "unique solution/no solution/rule holds" after testing only a few small values;
          - Replacing strict elimination with intuition, such as "grows faster/unlikely";
          - Finding only partial solutions without proving there are no more.
     - Why it might still be correct:
       - The actual solutions do fall within the tested small range or are indeed limited to those found; or although the pattern is
          wrong, the count within the given range happens to match the correct pattern (density/period coincidence).

2. Outcome Irrelevance (rounding/missing multiplication/sign errors in irrelevant parts, or double errors canceling out)
     - Typical symptoms:
       - Rounding too early in the process, but the final result is only reported to the tenths place, so the error does not amplify;
       - Missing the imaginary part/coefficient/negative sign, but only taking the real part/absolute value or m+n (order irrelevant)
          in the end;
       - Introducing an extra denominator first, then "forgetting" it later, which happens to cancel the error; two miscalculated
          numbers add up to the correct value.
     - Why it might still be correct:
       - The quantity sought in the problem is insensitive to the error (only depends on the real part/absolute value/last digit/
          modulus), or the error is swallowed by rounding in the end;
          - Two independent errors accidentally cancel each other out (negative times negative makes positive).

3. Neglected Operational Preconditions (domain/reversibility conditions/boundary points, but coincidentally not affecting)
     - Typical symptoms:
          - Directly canceling/dividing by a variable without first stating that the variable is not zero;
          - Converting log(x²) to 2log x without first restricting x>0;
       - Simplifying a fractional equation without first stating that the denominator is not zero; ignoring whether boundary points
          should be included.
     - Why it might still be correct:
       - The calculated value happens to satisfy the (unwritten) domain or reversibility conditions, thus no extraneous or missing
          roots are produced;
          - Other terms in the problem automatically restrict the domain (e.g., the equation already contains log x, implicitly
          requiring x>0).

4. Unverified Assumptions (unproven structural assumptions/misapplying theorems but hitting equality conditions or special cases)
     - Typical symptoms:
       - Directly assuming "the function must be linear", "extremum occurs when variables are equal", "a trapezoid has maximum area
          as a rectangle", "choosing a seemingly reasonable parameter value r=7", etc.;
          - Misapplying theorems (applying quadrilateral properties to hexagons, misusing properties like radical axes/exterior
          angles, etc.).
     - Why it might still be correct:
       - The guessed structure happens to be the equality condition or a hidden special property in the problem (such as symmetry,
          equality condition of Cauchy's inequality, special cases in circle geometry), thus the conclusion is correct;
          - The misapplied theorem still holds as a "numerical equality" in this special case, or is equivalent to another correct
          property.

5. Numerical Coincidence (the problem-solving process is completely different from the correct method and logically invalid, but the
final answer is correct due to numerical coincidence)
     - Typical symptoms:
       - Using wrong logic and calculations to get an incorrect probability of 9/20, while the correct probability is 7/22. But the
          problem asks for m+n, and coincidentally 9+20=29 and 7+22=29, resulting in the same answer;
       - Constructing an incorrect list of numbers that completely fails to meet the problem's conditions, but the square sum of this
          wrong list happens to equal that of the correct list;
       - Deriving an incorrect pattern of winning/losing conditions based on wrong game analysis, but within the given numerical
         range, the number of numbers satisfying this wrong pattern is exactly the same as those satisfying the correct pattern.
     - Why it might still be correct:
          - Coincidence.

6. Miracle Steps (the model's solution contains invalid steps, but suddenly arrives at the correct answer)
     - Typical symptoms:
       - The model lists a completely wrong equation "a + b + c + d - 437 - 2*234 - 3x = 3600", solves x=-827 (wrong answer) according to
          this equation, but the next step directly gives x=73 (correct answer);
       - The model provides a series of wrong ideas and steps, but suddenly lists a correct equation/inequality in an incomprehensible
          way.

7. Other
```

Please use Chinese and output the results in the following format:

**Are there errors or imprecise points in the problem-solving process:**
Yes / No
**If there are problems, why the wrong process led to the correct answer:**
(This item can be omitted if there are no errors)
- Error type
- Explanation
- Final result: [1-7] (e.g., [1], [2,3])

---

## Prompt 2: Rubric Generation

**Role:** You are an experienced math competition coach and problem-setter, an expert in the logical structure of mathematical proofs. Your task is not to solve math problems, but to design a rigorous, universal, and actionable scoring framework for evaluating solution processes.

- Your output should only be the Grading Rubric (i.e. Detailed Scoring Rubric & Coach's Guide), with no other content.
- The total score is 10 points.

**Example Problem:** Given four points in a Cartesian coordinate system: A(0, 0), B(4, 0), C(4, 4), and D(0, 4). Prove that quadrilateral ABCD is a square.

**Guiding Principles:**

1. **Method-Agnostic:** This rubric must be able to fairly evaluate all logically correct solution methods, whether they use side lengths, angles, or diagonals. **Strictly prohibit** creating separate criteria for specific methods (e.g., "side-length method," "diagonal method").

2. **Structure-Based:** The core of the scoring should be based on the universal structure of a proof, namely: "identifying key properties," "calculation and derivation," "logical linkage," and "final conclusion."

3. **Actionable Criteria:** The scoring criteria must be specific, observable actions, not abstract descriptions.
   - **Forbidden terms:** "accuracy," "rigor," "clear thinking," "fluent expression."
   - **Encouraged phrases:** "Correctly writes the distance formula," "Explicitly states that the slopes of two segments are negative reciprocals," "Concludes C based on previously proven properties A and B," "Completely states the theorem for identifying a square."

**Rubric Framework:**

Please break down the scoring rubric into the following sections and assign appropriate points to each (the total score is set to 10 points).

1. **Target Identification & Strategy Statement - [e.g., 1 point]**
   - Scoring Point: The student clearly identifies the objective (to prove it's a square) and articulates the set of mathematical properties their chosen strategy relies on.
   - Example: "To prove it's a square, I will show that all four sides are equal and one interior angle is a right angle." or "I will prove it's a square by showing its diagonals are perpendicular, bisect each other, and are equal in length."

2. **Calculation & Verification of Properties - [e.g., 6 points]**
   - This is the core of the rubric. The student must use calculations to verify **all** key properties required by their chosen strategy. This section is scored based on "properties," and regardless of the method, the student must prove a set of **sufficient conditions**.
   - **Scoring Points (detailed by property):**
     - **Proof of Property 1:** [e.g., Equal side lengths]
       * Correctly applies the necessary formula (e.g., distance formula).
       * Calculation is free of errors, and lengths of all sides are found.
       * Reaches an intermediate conclusion of equal side lengths (e.g., AB=BC=CD=DA=4).
     - **Proof of Property 2:** [e.g., Perpendicular adjacent sides or perpendicular diagonals]
       * Correctly applies the necessary method (e.g., slope calculation, vector dot product).
       * Calculation is free of errors, leading to the conclusion of perpendicularity.
     - **Proof of Property 3:** [e.g., Equal diagonals or diagonals that bisect each other]
       * ... (and so on)
   - **Note:** When scoring, check if the student has completely proven a **full set** of sufficient conditions for their chosen strategy. For example, only proving four equal sides (which could be a rhombus) does not earn full points for this section.

3. **Logical Synthesis & Final Conclusion - [e.g., 3 points]**
   - **Scoring Point 1 - Citing the Justification:** The student explicitly cites a definition or theorem that links the verified properties to the final conclusion. Example: "Because quadrilateral ABCD has four equal sides and one right angle, it is a square."
   - **Scoring Point 2 - Final Statement:** Provides a clear, conclusive statement. Example: "Therefore, quadrilateral ABCD is a square. Q.E.D."
   - **Scoring Point 3 - Logical Integrity:** The proof is free of logical gaps. For example, the student doesn't just calculate lengths and slopes and then jump to the conclusion without stating what those numbers mean (e.g., "sides are equal" or "sides are perpendicular").

**Output Format:** Please present the final rubric in a clear table format, including "Scoring Item," "Specific Criteria," and "Score." Here is an example:

**Core Task:** Design a detailed Scoring Rubric for the following math problem. The total score is 10 points.

[Insert the specific math problem here]

**Prompt 3: Scoring Data Generation**

For each Question and Answer pair, please evaluate according to the given scoring criteria.

- Your output must include two sections: Analysis and Summary.

- In your analysis, always place the score below your reasoning using the following format:

```
- Reason: [Insert your explanation here]
- Score: X
```

- In the summary, provide your assessment using the following format:

```
Final Scoring Summary:

Scoring Criterion 1 (Brief description of the criterion):
(Reason: ...) X points / Y points

Scoring Criterion 2 (Brief description of the criterion):
(Reason: ...) X points / Y points

...

Total Score: [N points / M points]
```

\# Question:
{INSERT QUESTION}

\# Scoring Criteria:
{INSERT CRITERIA}

\# Answer:
{INSERT MODEL_ANSWER}

**Prompt 4: Rubric Model Scoring Template**

You are a grade teacher. Based on the following information, please evaluate the student's submitted answer.

The total score is out of 10 points.

Carefully read the question, the scoring criteria, and the submitted answer. Then give a fair and reasonable score according to the grading criteria.

Here is the information:

Question:
{INSERT QUESTION}

Scoring Criteria:
{INSERT CRITERIA}

Student's Answer:
{INSERT MODEL_ANSWER}

# B CASE STUDY

> ## Question
>
> We are told that the process $\mathcal{P}$ consists of:
>   1. Reflect the point in the $x$-axis.
>   2. Translate the resulting point 2 units upwards.
>   3. Reflect the resulting point in the $y$-axis.
> We are given $S_6 = (-7, -1)$ and asked to find $S_0$ after reversing $\mathcal{P}$ six times.

> ## Baseline's Solution
>
> ### Solution
>
> To determine the coordinates of the point $S_0$, we need to reverse the steps of the process $\mathcal{P}$ applied six times to obtain the point $S_6(-7, -1)$. Let's break down the process step by step.
>
> **Step-by-Step Reasoning:**
>
> 1. **Identify the final point $S_6$:**
>    $S_6 = (-7, -1)$
>
> 2. **Reverse the sixth application of $\mathcal{P}$:**
>    - $S_6$ is the result of applying $\mathcal{P}$ to $S_5$.
>    - To find $S_5$, we need to reverse the steps of $\mathcal{P}$:
>      (a) Step 3: Reflect $S_6$ in the $y$-axis. This means $S_5$ has the same $y$-coordinate as $S_6$ but the opposite $x$-coordinate. $S_5 = (7, -1)$
>      (b) Step 2: Translate $S_5$ 2 units downwards. This means $S_4$ has the same $x$-coordinate as $S_5$ but the $y$-coordinate is 2 units less. $S_4 = (7, -3)$
>      (c) Step 1: Reflect $S_4$ in the $x$-axis. This means $S_3$ has the same $x$-coordinate as $S_4$ but the opposite $y$-coordinate. $S_3 = (7, 3)$
>
> 3. **Reverse the fifth application of $\mathcal{P}$:**
>    - $S_3$ is the result of applying $\mathcal{P}$ to $S_2$.
>    - To find $S_2$, we need to reverse the steps of $\mathcal{P}$:
>      (a) Step 3: Reflect $S_3$ in the $y$-axis. This means $S_2$ has the same $y$-coordinate as $S_3$ but the opposite $x$-coordinate. $S_2 = (-7, 3)$
>      (b) Step 2: Translate $S_2$ 2 units downwards. This means $S_1$ has the same $x$-coordinate as $S_2$ but the $y$-coordinate is 2 units less. $S_1 = (-7, 1)$
>      (c) Step 1: Reflect $S_1$ in the $x$-axis. This means $S_0$ has the same $x$-coordinate as $S_1$ but the opposite $y$-coordinate. $S_0 = (-7, -1)$
>
> Thus, the coordinates of the point $S_0$ are $\boxed{(-7, -1)}$.
>
> ### False Positive Analysis
>
> The student's problem-solving method contained a fundamental logical error. The student attempted to deduce $S_0$ from $S_6$ by reversing the transformation process $\mathcal{P}$. A single inverse process $\mathcal{P}^{-1}$ indeed requires performing the opposite operations in reverse order (first reflect across the $y$-axis, then translate downward by 2, and finally reflect across the $x$-axis). However, the student incorrectly assumed that **each individual sub-step** in the inverse process corresponded to one backward step in the sequence. For example, in the process of solving for $S_5$ from $S_6$, the student recorded the result after the first inverse operation (reflection across the $y$-axis) as $S_5$, the result after the second inverse operation (translation downward) as $S_4$, and the result after the third inverse operation (reflection across the $x$-axis) as $S_3$. This is a serious logical flaw. The correct approach is to apply **all three inverse operations** to $S_6$; the final result, after completing all steps, is $S_5$. According to the student's erroneous logic, they actually performed two complete inverse processes:
>
>   1. In the first complete inverse operation, they believed they had derived $S_3$ from $S_6$.
>
>   2. In the second complete inverse operation, they believed they had derived $S_0$ from $S_3$.
>
> Thus, the student effectively computed $(\mathcal{P}^{-1})^2(S_6)$, which is the coordinate of $S_4$, but they labeled it as $S_0$.
>
> The fact that the final answer was correct is purely coincidental. The transformation $\mathcal{P}$ has a special property: it has a period of 2. That is, applying $\mathcal{P}$ twice to any point returns the point to its original position (i.e., $\mathcal{P}(\mathcal{P}(S)) = S$). This means the sequence $S_0, S_1, S_2, S_3, S_4, S_5, S_6, \ldots$ is periodic with period 2, specifically $S_0 = S_2 = S_4 = S_6$ and $S_1 = S_3 = S_5$.
>
> Due to this periodicity, the coordinates of $S_4$ are exactly the same as those of $S_0$. The student mistakenly computed the value of $S_4$, but because $S_4 = S_0$, they arrived at the correct answer for $S_0$ purely by

coincidence. The student's problem-solving process showed no awareness of or reliance on this periodicity; the correctness of the final answer was entirely accidental.

## Ours Solution

### Solution

Alright, I have this problem about a point moving through a series of transformations, and I need to find the original point based on the final position after multiple applications of a process called $\mathcal{P}$. Let's break it down step by step.

**Understanding the Problem**

First, let's understand what's given:
1. **Initial Point**: $(3, 5)$
2. **Process $\mathcal{P}$**:
    - **Step 1**: Reflect the point in the $x$-axis.
    - **Step 2**: Translate the resulting point 2 units upwards.
    - **Step 3**: Reflect the resulting point in the $y$-axis.
3. **Applications**:
    - Start with point $S_0$.
    - Apply $\mathcal{P}$ to $S_0$ to get $S_1$.
    - Continue applying $\mathcal{P}$ four more times to get $S_6$.
4. **Final Point**: $S_6(-7, -1)$
5. **Goal**: Find the coordinates of $S_0$.

**Breaking Down the Process $\mathcal{P}$**

Let's understand each step of $\mathcal{P}$ and how it transforms a point $(x, y)$.
1. **Step 1: Reflect in the $x$-axis**
    - Reflecting a point in the $x$-axis changes the sign of the $y$-coordinate.
    - So, $(x, y)$ becomes $(x, -y)$.
2. **Step 2: Translate 2 units upwards**
    - Translating a point upwards increases the $y$-coordinate by 2.
    - So, $(x, -y)$ becomes $(x, -y + 2)$.
3. **Step 3: Reflect in the $y$-axis**
    - Reflecting a point in the $y$-axis changes the sign of the $x$-coordinate.
    - So, $(x, -y + 2)$ becomes $(-x, -y + 2)$.

Therefore, the entire process $\mathcal{P}$ transforms $(x, y)$ to $(-x, -y + 2)$.

**Applying $\mathcal{P}$ Multiple Times**

We start with $S_0$ and apply $\mathcal{P}$ six times to get $S_6$. Let's denote the transformation as $T$, so:
$$T(x, y) = (-x, -y + 2)$$
Applying $T$ multiple times:
$$T^n(x, y) = T(T^{n-1}(x, y))$$
We need to find a pattern or a formula for $T^n(x, y)$. Let's compute the first few applications to see if a pattern emerges.

1. $T^1(x, y)$:
$$T^1(x, y) = (-x, -y + 2)$$

2. $T^2(x, y)$:
$$\begin{aligned}
T^2(x, y) &= T(T^1(x, y)) \\
&= T(-x, -y + 2) \\
&= (-(-x), -(-y + 2) + 2) \\
&= (x, y - 2 + 2) \\
&= (x, y)
\end{aligned}$$

Wait, $T^2(x, y) = (x, y)$. That's interesting! Applying $T$ twice brings us back to the original point.

3. $T^3(x, y)$:
$$T^3(x, y) = T(T^2(x, y)) = T(x, y) = (-x, -y + 2)$$
So, $T^3(x, y) = T^1(x, y)$.

4. $T^4(x, y)$:
$$T^4(x, y) = T(T^3(x, y)) = T(-x, -y + 2) = (x, y)$$
Again, $T^4(x, y) = (x, y)$.

From these computations, it seems that applying $T$ an even number of times returns the point to its original position, and applying $T$ an odd number of times gives the same result as applying $T$ once.

**Generalizing the Transformation**

Given that $T^2(x, y) = (x, y)$, we can see that:

- If $n$ is even, $T^n(x, y) = (x, y)$.
- If $n$ is odd, $T^n(x, y) = T^1(x, y) = (-x, -y + 2)$.

Since we're applying $T$ six times ($n = 6$), which is even, we have:
$$T^6(x, y) = (x, y)$$
But we know that $T^6(S_0) = S_6 = (-7, -1)$. Therefore:
$$(x, y) = (-7, -1)$$
So, $S_0 = (-7, -1)$.

**Verification**

Let's verify this by applying $\mathcal{P}$ six times starting with $S_0 = (-7, -1)$ and see if we get $S_6 = (-7, -1)$.

1. $S_0 = (-7, -1)$
2. $S_1 = T(S_0) = (-(-7), -(-1) + 2) = (7, 1 + 2) = (7, 3)$
3. $S_2 = T(S_1) = (-7, -3 + 2) = (-7, -1)$
4. $S_3 = T(S_2) = (7, 1)$
5. $S_4 = T(S_3) = (-7, -1)$
6. $S_5 = T(S_4) = (7, 3)$
7. $S_6 = T(S_5) = (-7, -1)$

Yes, after six applications, we indeed get $S_6 = (-7, -1)$. This confirms that our solution is correct.

**Final Answer**

The coordinates of the point $S_0$ are $\boxed{(-7, -1)}$.

**False Positive Analysis**

The student's overall problem-solving approach and core derivation process are completely correct. They correctly derived the algebraic expression for the single transformation $\mathcal{P}$ as $T(x, y) = (-x, -y + 2)$, and found through calculation that the second iteration of the transformation, $T^2(x, y)$, is the identity transformation, i.e., $T^2(x, y) = (x, y)$. Based on this key discovery, they concluded that "applying the transformation an even number of times will return the point to its original position," and therefore $S_6 = T^6(S_0) = S_0$. This chain of reasoning is rigorous and flawless, directly leading to the correct answer $S_0 = S_6 = (-7, -1)$.

The error occurred in the final "Verification" step, which was intended as a post-solution check. In computing the verification sequence, the student made a slip or calculation error:

- They correctly computed $S_2 = (-7, -1)$.
- Next, when calculating $S_3$, they wrote $S_3 = T(S_2) = (7, 1)$.
- The correct computation should be $S_3 = T(-7, -1) = (-(-7), -(-1) + 2) = (7, 1 + 2) = (7, 3)$.

# C SUPPLEMENT

## C.1 FALSE POSITIVE ANALYSIS PROCEDURE

There are four stages for analyzing false positive modes:

*Stage 1: Data Preparation.* We assemble a dataset of 680 samples, comprising 170 distinct questions (30 from AIME2024 + 50 from MATH500 + 40 from AMC2023 + 50 from Olympiad), each with four unique model responses. All markdown and mathematical formulas have been converted into an easily readable format.

*Stage 2: Initial Mode Discovery.* We use Gemini-2.5-Pro for an automated review to generate a preliminary taxonomy of "false positive modes." The model is prompted with each question, a reference solution, and the model's response, and is instructed to report on (1) any reasoning errors and (2) how flawed reasoning can still yield a correct answer. These reports are then aggregated and synthesized by the model into the initial taxonomy.

*Stage 3: Expert Review.* In the third stage, we conduct a human validation of these modes. Four annotators, all holding undergraduate degrees with substantial training in advanced mathematics, evaluate each sample. They are equipped with tools like Google Search and large models and are instructed to discard any samples beyond their expertise. For each sample, they determine if it is a false positive and, if so, classify it using our preliminary taxonomy or label it as "Other" with a detailed explanation.

*Stage 4: Synthesis and Analysis.* In the final stage, we refine the taxonomy by incorporating the "Other" categories identified by human annotators. Using this final, human-validated framework, we perform a quantitative analysis to measure the frequency of each false positive mode, revealing the model's prevalent reasoning flaws.

## C.2 DISCARDED QUESTION

> **Question**
>
> In the circle with center $Q$, radii $AQ$ and $BQ$ form a right angle. The two smaller regions are tangent semicircles, as shown. The radius of the circle with center $Q$ is 14 inches. What is the radius of the smaller semicircle? Express your answer as a common fraction.

## C.3 EXPERIMENTAL DETAILS FOR STATE-OF-THE-ART MODEL EVALUATION

*Models and Generation.* We evaluated four leading models: GPT-5-thinking, o4-mini, Gemini-2.5-Pro, and Claude-4-Sonnet-thinking. We employ Gemini-2.5-Pro (version 0605). For the other models, namely o4-mini, GPT-5, and Claude-4-Sonnet, we utilize their latest versions available as of September 2025. For each question in the challenge set, we generated a single response from each model ($n = 1$). To encourage more detailed reasoning, we set the reasoning effort parameter to 'high' for both GPT-5-thinking and o4-mini.

*Evaluation Protocol.* All generated responses were manually evaluated by human annotators.

*Additional notes.* During annotation, we noted that o4-mini exhibited a strong tendency to provide overly concise or truncated reasoning steps. This brevity sometimes made it challenging to fully assess the validity of its solution path and may contribute to its higher observed false positive rate, as critical (and potentially erroneous) intermediate steps might be omitted.

*Qualitative Examples.* For qualitative insights, several examples of questions from our challenge set that frequently induced false positives across the evaluated models are presented below:

---

***Question 1.*** Rectangles $ABCD$ and $EFGH$ are drawn such that $D, E, C, F$ are collinear. Also, $A, D, H, G$ all lie on a circle. If $BC = 16, AB = 107, FG = 17$, and $EF = 184$, what is the length of $CE$?

***Failure***: All models overlook the possible permutations of $D, E, C, F$.

---

***Question 2.*** How many ordered pairs of positive real numbers $(a, b)$ satisfy the equation

$$(1 + 2a)(2 + 2b)(2a + b) = 32ab?$$

***Failure***: Claude-4-Sonnet directly identified the correct (a,b) pair through trial, then reported unsuccessful attempts with alternative answers, and subsequently claimed that only one such pair satisfies the requirements. GPT-5 ignored the case of a zero denominator during its simplification process. o4-mini made an error in its variable substitution step.

---

***Question 3.*** Rows 1, 2, 3, 4, and 5 of a triangular array of integers are shown below.
1
1 1
1 3 1
1 5 5 1
1 7 11 7 1
Each row after the first row is formed by placing a 1 at each end of the row, and each interior entry is 1 greater than the sum of the two numbers diagonally above it in the previous row. What is the units digits of the sum of the 2023 numbers in the 2023rd row?

***Failure***: Gemini-2.5-Pro and Claude-4-Sonnet, through enumeration, discovered an important function $U(\cdot)$ in solving the problem have: $U(21) = U(1)$. Without providing proof, they directly claimed the existence of periodicity.

---

## C.4 RRM TRAINING DETAILS

We fine-tune the Qwen3-4B-Base model as our policy model using PPO. The training is guided by a reward function, which is calculated based on the L1 distance between the predicted score ($\text{Score}_{\text{pred}}$) from our reward model and the target score ($\text{Score}_{\text{target}}$):

$$\text{Reward} = 1 - \left| \frac{\text{Score}_{\text{pred}} - \text{Score}_{\text{target}}}{10} \right|.$$

The PPO training is configured with the following hyperparameters: a maximum prompt length of 10000, a maximum sequence length of 2048, a batch size of 128, and a rollout size of 8. We use the Adam optimizer with a learning rate of $5 \times 10^{-7}$ and a generation temperature of 1.0. The model is trained for 400 steps.

## C.5 THE SCORING ACCURACY AND STABILITY OF GEMINI-2.5-PRO.

When using Gemini-2.5-Pro for scoring, we set the temperature to 1.0, perform repeated sampling five times, and calculate the difference between the highest score and the lowest score among these five runs. As can be seen from the Figure 8, Gemini-2.5-Pro demonstrates good stability despite minor fluctuations.

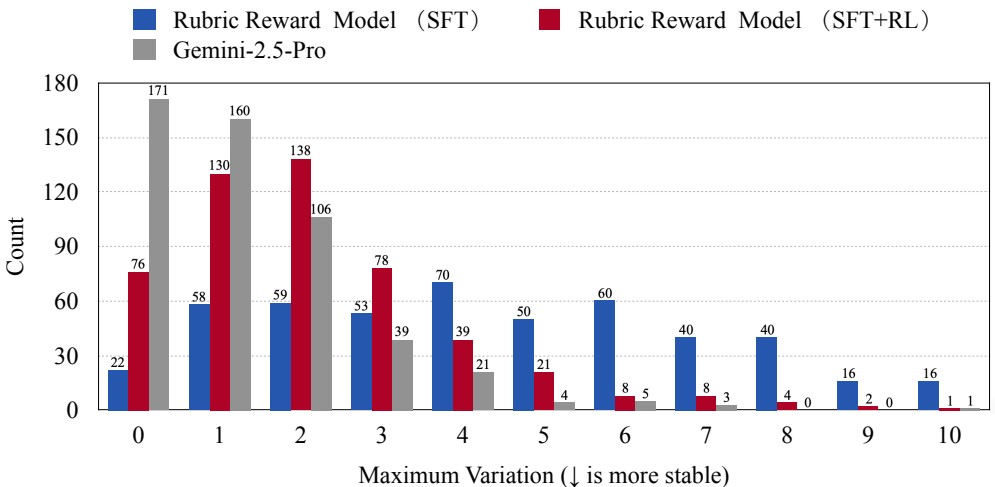

Figure 8: The scoring stability of Gemini-2.5-Pro.

Table 3: Manual evaluation of the accuracy of Gemini's scoring according to the rubric.

| Gemini Rubric Scoring | Too high | Too low | Accurate |
|:---:|:---:|:---:|:---:|
| *Count* | 12 | 7 | 1301 |

## C.6  DETAILS FOR MAIN EXPERIMENTS

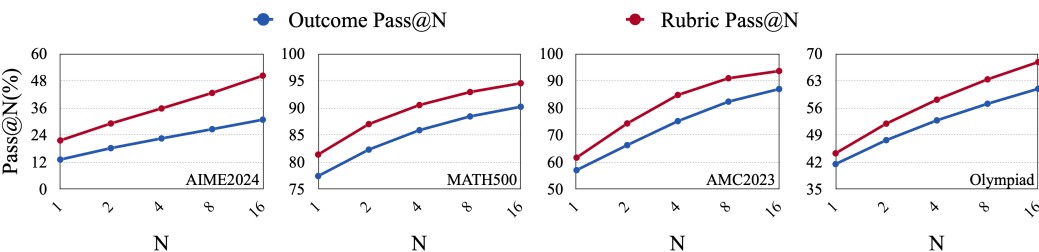

Figure 9: Qwen3-4B's Pass@N results on the full dataset.

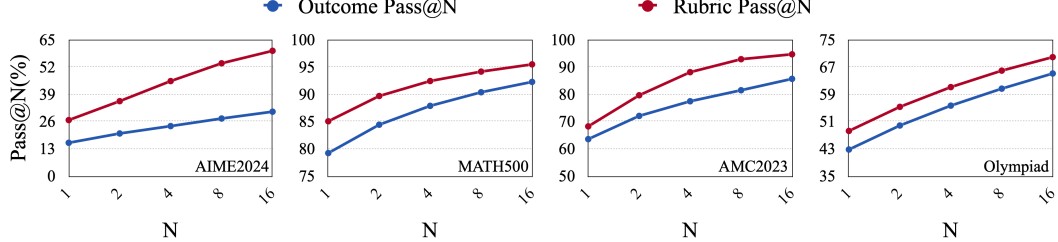

Figure 10: Qwen-8B's Pass@N results on the full dataset.

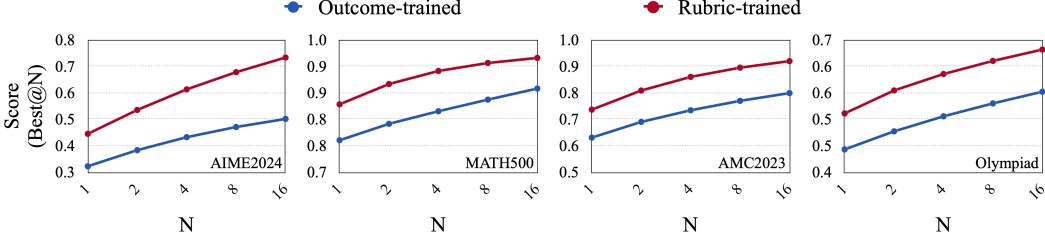

Figure 11: Qwen3-4B's Gemini scoring results on the full dataset.

All our training and inference were conducted on a server with 8 NVIDIA A800-80G GPUs. During evaluation, we set the temperature to 1.0, the maximum generation length to 16,000 tokens, and used the prompt:

*Please reason step by step, and put your final answer within* \boxed{}.

To evaluate *Pass@N*, we generate 2N candidate solutions for each problem instance.

**Evaluation on full datasets and the Qwen3-8B.**   In our main experiments, due to computational cost considerations, we randomly selected a subset of 50 samples from MATH500 (500 samples) and Olympiad (675 samples) for evaluation. We additionally conducted experiments on the full datasets (32 runs), and the results are presented in Figure 9 and 10. The overall trends and conclusions remain consistent with those observed on the subset.

**Comparison of the scores assigned by Gemini-2.5-Pro to our model and the baseline models.** As a supplementary result, Figure 11 presents the outcomes of using Gemini-2.5-Pro to generate a rubric on the test set and to score the responses of both models.

In our distributional analysis of error cases (Section 5.3), we focus on instances that were not assigned a perfect score by Gemini-2.5-Pro. The rationale is that false-positive samples with a perfect Gemini grade represent cases where the rubric reward is inherently unable to address the issue. In contrast, our error analysis aims to examine cases in which the rubric reward could potentially play a role.

## D HUMAN EVALUATION

### D.1 GEMINI-2.5-PRO AS A FALSE POSITIVE JUDGER: RELIABILITY ASSESSMENT

Table 4: Confusion matrix comparing false positives identified by human and by Gemini.

| Samples (Overall) | | Gemini TP | Gemini FP |
|---|---|---|---|
| Human | TP | 462 | 93 |
| | FP | 9 | 295 |

| Samples (Rubric) | | Gemini TP | Gemini FP |
|---|---|---|---|
| Human | TP | 252 | 52 |
| | FP | 1 | 152 |

| Samples (Outcome) | | Gemini TP | Gemini FP |
|---|---|---|---|
| Human | TP | 210 | 41 |
| | FP | 8 | 144 |

Table 5: Confusion matrix on different datasets.

| Samples (AIME) | | Gemini TP | Gemini FP |
|---|---|---|---|
| Human | TP | 34 | 8 |
| | FP | 1 | 28 |

| Samples (AMC) | | Gemini TP | Gemini FP |
|---|---|---|---|
| Human | TP | 112 | 17 |
| | FP | 2 | 105 |

| Samples (MATH) | | Gemini TP | Gemini FP |
|---|---|---|---|
| Human | TP | 222 | 50 |
| | FP | 0 | 44 |

| Samples (Olympiad) | | Gemini TP | Gemini FP |
|---|---|---|---|
| Human | TP | 94 | 18 |
| | FP | 6 | 118 |

Table 6: The proportion of questions for which the model and human false positive evaluations are identical across all responses to that question.

| Human-Gemini Consistency | Qwen3-Outcome (4 resp. per query) | Qwen3-Rubric (4 resp. per query) | Overall (8 resp. per query) |
|---|---|---|---|
| *Ratio* | 92/121 | 109/139 | 97/141 |

**Agreement with human experts.** We quantify Gemini-2.5-Pro's reliability by conducting extensive human evaluation. As shown in Table 4, Gemini attains high precision (98.1%) and reasonable recall (83.2%) against human labels, yielding an overall F1 score of 0.90 and an agreement rate of 88.1%. These results confirm that Gemini correctly flags almost all human-identified false positives and makes very few spurious accusations.

**No preference toward rubric/outcome-trained outputs.** Empirically, Gemini exhibits comparable behavior on rubric-trained and outcome-trained responses. From Table 4:

- Rubric-trained subset: precision 99.6%, recall 82.9%, agreement 88.4%.
- Outcome-trained subset: precision 96.3%, recall 83.6%, agreement 87.9%.

The near-identical recalls (82.9% vs 83.6%) and close agreement rates (88.4% vs 87.9%) show no systematic advantage for rubric-trained outputs; if anything, the tiny precision difference reflects fewer false alarms on that subset, not preferential scoring.

**Consistency across datasets.** The performance is stable across datasets (Table 5): F1 ranges from 0.88 (AIME) to 0.92 (AMC), with precision consistently $\geq 0.94$. This robustness suggests that Gemini's accuracy is not confined to a particular problem source or difficulty level.

**Agreement at question level.** We also assess whether Gemini-2.5-Pro and human annotators agree *across all responses* to the same prompt. Complete question-level agreement holds for 76.0% of questions in the outcome-trained setting, 78.4% in the rubric-trained setting, and 68.8% overall (Table 6). The similar agreement rates for rubric- and outcome-trained models indicate that Gemini does not systematically favor one training method over the other.

Given its high precision, stable cross-dataset performance, and absence of bias toward our method, we use Gemini-2.5-Pro as a scalable, automatic false-positive judge for the remainder of our analysis.

