# OpenReview forum: "Curing "Miracle Steps'' in LLM Math Reasoning with Rubric Rewards"
_ICLR.cc/2026/Conference — ICLR 2026 Conference Withdrawn Submission_

### Official Review · Reviewer_7aZN · 2025-10-15

**Soundness:** 3
**Presentation:** 2
**Contribution:** 3
**Rating:** 4
**Confidence:** 4

**Summary:**

This paper introduces rubric reward models (RRMs) with the aim to train LLMs to produce better reasoning that more accurately derives a final answer. To motivate this, the authors perform experiments showing that LLMs generate a lot of false positives: reasonings that lead to the correct final answer, but contain various mistakes along the way. The RRM is trained by distilling from Gemini-2.5-Pro with some RL on top of it. The RRM is then used to perform PPO on an LLM, significantly improving the LLMs performance and reducing the false positives compared to a baseline.

**Strengths:**

- The paper idea is interesting and well-executed.
- Section 3 is appreciated and the correct experiments are performed.
- The writing is clear, and many details are included.

**Weaknesses:**

I think the paper definitely has potential, but some aspects raise questions that need to be resolved or elaborated upon before I feel comfortable to give an accept rating. I have split my concerns in weaknesses and remarks, the latter only influencing my score in minor ways.

**Weaknesses**
- Section 3.2 presents an interesting experiments that should have one of two additional conclusions, neither is followed by the authors. Either:
  - The benchmarks should be considered completely contaminated and not used anymore for evaluation. Unfortunately, these are the only benchmarks included in the analysis. The authors are strongly encouraged to present evaluation results on more recent benchmarks such as AIME 2025 or other datasets from 2025.
  - The experiment does not actually measure contamination, and the model can guess correctly because: (1) the question is so easy it does not need reasoning, (2) the answer is "guessable" because it is for instance a low number. This would imply the conclusions drawn by the authors are incorrect, as the answers are not an indication of memorisation.
Likely the effect is a combination of both, but the authors should devise experiments to tell how large the effect of each is.
- It is unclear why PPO is preferred over GRPO (or variants of it), especially since GRPO is much more common these days to finetune LLMs for mathematical tasks. Experiments with GRPO need to be performed.
- RL is notoriously unstable. Differences in performances could just as well be caused by one training run failing or degrading at the step RL training is finished. The authors should repeat the same experiment several times to ensure that the performance difference is not due to these instabilities.
- The authors neglect to cite any works that evaluate proofs produced by LLMs. Particularly relevant to this work is [1]: this work already notes that outcome-based evaluation is not sufficient and there is a significant discrepancy between final-answer accuracy and being able to provide correct proofs ("false positives" in the authors' words). The categorization presented there has also an overlap with the presented categorization here (there are differences, but not sure if one is better than the other).
- There are also some baselines missing in the comparison. The authors should compare against using a PRM, ORM, or a prompted LLM (without rubrics) as reward model, not only the final-answer outcome reward. Based on 4.1, they are likely to perform a bit worse,  but it would still be interesting.

**Remarks**
- Experiments are performed on non-reasoning models (max 4k tokens). This limits the applicability, although it is somewhat expected that the results generalize to reasoning models.
- If I am not mistaken, ReasonFluxPRM is specifically finetuned to give rewards for reasoning traces (the <think></think> phase in reasoning models), but the evaluation in 4.1 does not take place on reasoning traces (so without <think></think> phase), but rather on the answers provided by models. The authors do mention in L282 the necessity to be able to handle "wait, ...", but it is unclear why this is necessary. At best, the authors should more clearly specify what type of data is used in this experiment. At worst, this becomes a weakness.
- It would be better to add specific numbers of average deviation from Gemini-2.5-Pro in addition to Fig. 5. e.g., deviation for RL is on average 1.1 and for SFT is 1.5.
- I am a bit worried about Figure 7. Usually, RL training for mathematical reasoning only increases performance because response length increases (maybe not causal, but there is a correlation). However, the baseline included in the paper shows no such pattern. How much does the model improve during its training run? It might just be learning formatting (which is usually already learned after 10 steps or so) and not much else.
- It should be mentioned in the main text that evaluation is only performed on a small subset of the benchmarks (as mentioned in the appendix).

[1] https://arxiv.org/pdf/2506.06877

**Questions:**

- For the results in Table 2, how can one tell the difference between a miracle step and something the model just neglected to mention in its final reply compared to its reasoning trace? The appendix mentioned that this definitely happened for o4-mini, and I am worried this happened on a larger scale, especially since it seems that the authors simply asked the question, not asking for rigorous reasoning or something like it. This might have caused models to skip steps even though they knew what the intermediate steps were.
- Why is the PRM saturation prone? (L301)

---

### Official Review · Reviewer_WhT6 · 2025-10-27

**Soundness:** 1
**Presentation:** 3
**Contribution:** 2
**Rating:** 2
**Confidence:** 4

**Summary:**

This paper aims to address the "false positive" reasoning when LLMs are applied to math reasoning, where the models provide the correct final answer but an unsound reasoning trace. The authors first highlight the importance of this issue by demonstrating that various LLMs, even the frontier ones, can exhibit such weaknesses. The authors then propose to mitigate this problem through RL training with a rubric reward model, which is distilled from Gemini-2.5-Pro. The empirical experiments show that the designed rubric reward model yield strong policies that are trained from Qwen3-4B-Base, compared to the outcome-reward-model-based baseline.

**Strengths:**

1. The core problem studied in this paper, the false negative reasoning of LLMs, is well-motivated by a detailed empirical analysis provided in the manuscript, which demonstrates the importance of addressing this issue.

2. In general, the paper is well-written and the presentation is clear.

**Weaknesses:**

1. In general, the empirical experiments presented in this work are not sufficient enough to support the center claim of this work that the proposed rubric-based reward model is superior than the baseline methods (outcome-based reward models, etc.) That said, some of the following concerns are due to the unclear details in this manuscript, which might be addressed with further clarification.

   (a) In Section 5.1, it is unclear how the outcome-based reward model, which is compared as the baseline, is constructed and trained. As the rubric reward model (RRM) proposed in this work is distilled from Gemini-2.5-Pro, it would only be a fair comparison if the baseline outcome-based reward model has undergone a similar training process using labels generated from Gemini-2.5-Pro.

   (b) Another important baseline that should be compared is a generative reward model/LLM-as-a-Judge that works similar as the RRM but without the specify rubrics in the prompt. Again, such a model can also be distilled from Gemini-2.5-Pro. This comparison is important since it clarifies how important the introduced rubrics are.

   (c) The experiments are conducted on only one base LLM, Qwen3-4B-Base. At least one more LLM should be considered as the base model/policy in RL to ensure the comprehensiveness and the robustness of the experiments.


2.  Some other important details also seem to be missing from the manuscript.

     (a) In Section 4.1, it is unclear whether the RRM is evaluated is the same RRM in 4.2 distilled from Gemini-2.5-Pro, or simply Qwen3-4B with a rubric-specific prompt. If it is the former, the comparison in section 4.1 appears unfair since the compared process reward model and the "false positive verifier" have not undergone such training processes based on Gemini-2.5-Pro.

     (b) The detailed information of the "expert" human evaluators discussed in the manuscript is lacking. How were they recruited and/or trained? What measures have taken to ensure the quality of the human annotations? What was the inter-annotator agreement among the human evaluators?

**Questions:**

Please see the "Weaknesses" section.

---

### Official Review · Reviewer_UuLc · 2025-10-28

**Soundness:** 2
**Presentation:** 4
**Contribution:** 3
**Rating:** 4
**Confidence:** 4

**Summary:**

The paper proposes training a Rubric-based Reward Model and using its reward score for RL post-training. The authors claim that their RRM-based training performs better on mathematical tasks, compared to using an Outcome-based reward score. The authors also provide a taxonomy of different types of “false positives” where a model’s final answer is correct but the reasoning is incorrect. The authors then compare the abilities of different techniques to identify the false positives.

Note: Even though the authors position away from PRMs, my personal opinion is that their proposed model is just a PRM that has been instruction tuned to look at the provided rubric.

**Strengths:**

1. The text is generally well written. The introduction and the motivation is clearly laid out. The experimental setup is easy to understand. The figures and their captions are descriptive.
2. The motivation of the paper is good, and the idea is good as well. “If a PRM cannot provide a good score for the step-by-step reasoning, maybe providing a concrete rubric given by a strong external model should help!”

While there are many weaknesses (see next section), the rubric-based reward scoring is an interesting idea to explore in the future and this paper would be a good addition to the community.

**Weaknesses:**

# Weaknesses
1. Weak connection between the “false positive phenomenon” and the improved performance with RRM. Specifically, the authors do not distinguish the effect of the false positives from the effect of “false negatives” and the limited training budget. The authors train with a 4k token budget, but evaluate with 16k token budget. For difficult questions, 4k might be too short for the model to get to the final answer. In which case, many correct attempts that did not lead to the correct final answer _yet_ (hence “false negatives”) may be penalized under outcome-based training. However, the authors seem to credit the deficiencies of the Outcome-based training solely to the false positives.

Note that Qwen3 technical report (Table 17) shows that Qwen3-4B-Base has a pass@64 accuracy (with 32k token budget) of 97% on MATH500, 73.8% on AIME24. These numbers are higher than the 90%, 40% for the Outcome-trained and 95%, 65% for RRM-trained checkpoints (Figure 6). The “drop” in performance compared to the base model likely comes from the fact that the Qwen3 report uses 32k tokens for evaluation, whereas the authors of this paper uses 16k for evaluation. It would be good to at least report on the evaluation scores of the base model they started training on (as another baseline), to see the absolute amount of improvement their RL training can bring.

2. The point above leads to the following point. Comparing against a single baseline of using a sparse output reward is unfair. The authors should at least compare against the baseline of training with the reward scores assigned by an external PRM. Although the authors make the claim that PRMs are bad at detecting false positives (Section 4.1), that doesn’t completely nullify the benefit of using a continuous reward, as opposed to the spare reward of 0/1. Especially if the discrepancy between the Outcome-trained and RRM-trained models is not solely due to the “false positives,” but also due to the limited training budget, the authors should consider showing the superiority of their proposed RRM over PRMs on assigning better training reward signals.

3. The proposed method seems expensive and relies heavily on large-scale LLMs to annotate data. The authors had to ask Gemini-2.5-Pro to
- annotate 680 solutions from Qwen3-4B-Outcome to create an initial taxonomy for the rubric. (which can potentially be amortized if this method is applied to a larger scale of data)
- annotate each problem in the training data (both for the RRM and for the policy model) with a rubric
- annotate each solution for the problems in the training data for the RRM.

One thing I am specifically worried about is the second point: since the RRM requires a rubric to assign a score, **every question in the training data for the policy model** needs to be annotated with a rubric. At that point, the cost of data annotation becomes comparable to distilling from Gemini-2.5-Pro. This defeats some benefits of the RL where data annotation costs are lighter than SFT and training can solely rely on the on-policy data generated by the policy model.

It would be helpful if the authors reported on the number of tokens necessary for this entire pipeline to assess the scalability of this method. how this would compare to e.g., directly generating solutions with Gemini-2.5-Pro and SFT on them.

4. The authors do not show the generalizability of their pipeline (more to following in the questions section)

With these weaknesses, it is hard to recommend an accept at this point of the project.


# Nit-picky
1. While a significant portion of the text revolves around the discussion of “Miracle Steps,” I really don’t agree that the main example on page 2 constitutes a “Miracle Step.” The model is randomly trying pairs of simple functions and gets one of the attempts correct. Every time the model tries a wrong attempt, it correctly recognizes that it is at a dead end and restarts the attempt with a different choice. This is very similar to how a human would approach these types of olympiad questions. If instead, the model was in the middle of one attempt, but suddenly pivots to the correct answer (without having finished the attempt), or if the model makes a wrong attempt but incorrectly assesses that a wrong attempt is in fact correct, but writes down the final answer as the correct answer, that would be considered “Miracle Steps.”

2. Appendix C.4 is missing the training details for the SFT. The relevant training scripts are also missing from the codebase.

3. The numbers in Table 2 (and also in lines 245-250) are not properly explained. “False positive rate” in statistics refers to FP / (FP + TN) but that doesn’t seem like the case here. The authors probably are referring to the “False discovery rate” (FP / (FP + TP))

**Questions:**

# Questions
1. Have you tried generating the rubric with other API models? Can your RRM accurately generate scores given a rubric generated by an API model, other than Gemini?
2. Does the RRM have to be the same model / of the same model family as the policy model?
3. Can the RRM generalize to out-of-distribution (difficulty-wise or at least a different data source) questions? Specifically, assume you want to train the policy model on a new math dataset that has been released after the RRM has been trained. How well can it generate reward scores for that dataset?
4. What is the exact set of datasets used to train the RRM? From my understanding, it is a part of the Polaris dataset where the Gemini-2.5-Pro was able to generate a rubric on, but it doesn’t seem to be explicitly mentioned.

---

### Official Review · Reviewer_yw3U · 2025-11-01

**Soundness:** 2
**Presentation:** 2
**Contribution:** 2
**Rating:** 2
**Confidence:** 3

**Summary:**

The paper examines how large language models often produce correct answers through flawed reasoning, termed false positives. A central failure mode, “Miracle Steps,” refers to abrupt jumps to the correct result without valid derivations, often linked to memorization. The authors develop a taxonomy of six types of such reasoning errors and show their presence even in advanced models like GPT-5 and Gemini-2.5-Pro. To address this, they propose the Rubric Reward Model (RRM), a process-oriented reward mechanism that scores the entire reasoning trajectory against problem-specific rubrics rather than only the final answer. RRM is evaluated through reinforcement learning experiments on four mathematical reasoning benchmarks, AIME2024, MATH500, AMC2023, and OlympiadBench, showing improved verified accuracy and a reduced incidence of Miracle Steps compared with outcome-based training.

**Strengths:**

- The paper addresses an important and relatively under-explored topic in post-training for reasoning.

- The proposed solution is straightforward and conceptually intuitive.

**Weaknesses:**

- The proposed approach relies on using a strong model to generate problem-specific rubrics and to serve as a judge, which introduces practical limitations for broader application.

- The comparison primarily focuses on outcome-based versus rubric-based rewards, but an important baseline, generative rewards, is missing. Such models might already mitigate false positives without the need for a sophisticated rubric design.

- The FP Verifier is a reasonable baseline; however, it is not specifically trained to detect false positives. An additional ablation where the FP Verifier is trained with reinforcement learning but without rubrics would help clarify whether the rubrics themselves are essential or if general RL-based training can achieve similar improvements.

**Questions:**

- The FP Verifier performs well on AIME and AMC but considerably worse on MATH500 and OlympiadBench. It would be valuable to analyze what factors contribute to this discrepancy.

- Following from this, it is possible that the Polaris dataset used for training overlaps with MATH500 or OlympiadBench. A decontamination analysis would help clarify whether the observed performance differences are influenced by data overlap rather than genuine generalization.

---

### Note · Authors · 2025-11-24

I have read and agree with the venue's withdrawal policy on behalf of myself and my co-authors.